# Halosmysin A, a Novel 14-Membered Macrodiolide Isolated from the Marine-Algae-Derived Fungus *Halosphaeriaceae* sp.

**DOI:** 10.3390/md18060320

**Published:** 2020-06-18

**Authors:** Takeshi Yamada, Haruka Kogure, Minami Kataoka, Takashi Kikuchi, Tomoya Hirano

**Affiliations:** Department of Medicinal Molecular Chemistry, Osaka University of Pharmaceutical Sciences, 4-20-1, Nasahara, Takatsuki, Osaka 569-1094, Japan; e13108@gap.oups.ac.jp (H.K.); e15329@gap.oups.ac.jp (M.K.); t.kikuchi@gly.oups.ac.jp (T.K.); hirano@gly.oups.ac.jp (T.H.)

**Keywords:** macrodiolide, 14-membered ring, *Halosphaeriaceae* sp., *Sargassum thunbergii*, colletodiol, cytotoxicity

## Abstract

Halosmysin A, a new 14-membered macrodiolide with an unprecedented skeleton, was isolated from the fungus *Halosphaeriaceae* sp. OUPS-135D-4, which, in turn, was obtained from the marine algae *Sargassum thunbergii.* The chemical structure of the macrodiolide was elucidated using 1D and 2D NMR, as well as high resolution fast atom bombardment mass (HRFABMS) spectral analysis. The absolute stereochemistry was determined via chemical derivatization and comparison with a known compound, (6*R*,11*R*,12*R*,14*R*)-colletodiol. Additionally, halosmysin A was shown to be very potent against murine P388 leukemia, human HL-60 leukemia, and murine L1210 leukemia cell lines, with IC_50_ values ranging from 2.2 ± 3.1 to 11.7 ± 2.8 μM.

## 1. Introduction

Marine-derived microorganisms are renowned for producing secondary metabolites, which have diverse structures and exhibit unexpected biological properties, making them excellent antitumor chemotherapy agents [1,2,3,4]. To date, our group has published numerous papers featuring exploratory research on marine-derived fungal metabolites [5]. As part of this ongoing study, the metabolites of *Halosphaeriaceae* sp.—a fungus which was separated from the marine algae *Sargassum thunbergii*—were isolated and characterized. In this study, a new 14-membered macrodiolide was designated as halosmysin A (**1**), together with the known colletodiol (**2**) [6] from this fungal strain, (Figure 1). Several related polyketide macrodiolides were isolated from diverse microorganisms, including **2**, colletoketol (grahamimysins A), colletoll, and colletallol from *Colletotrichum capsici* [6]; grahamimysins A_1_ and B from *Cytospora* sp. [7,8,9]; 9,10-dihydrocolletodiol from *Varicosporina ramulosa* [10]; clonostachydiol from *Clonostachys cylindrospora* [11]; *Gliocladium* sp. [12] and *Xylaria* sp. [13,14]; 4-keto-clonostachydiol from *Gliocladium* sp. [12,15]; cordybislactone from *Cordyceps* sp. [13]; and acremonol and acremodiol from an unidentified *Acremoniu*–like anamorphic fungus [16]. These macrodiolides exhibited diverse activities such as antimicrobial, antiosteoporosis, tyrosine kinase inhibition, anthelmintic, and cytotoxicity. Recently, other members of this family, namely acaulones, were isolated from the fungus *Acaulium* sp.; these acaulones possessed a new skeleton or functional group with the olefin moiety in the macrodiolide [17]. The stereostructures of these compounds were established by applying the several rules of chirality [6,10], X-ray crystallographic analysis [18], and asymmetric total synthesis [19,20,21,22,23]. After revisions were conducted [10,12,13,15], it was confirmed that these macrodiolides consisted of two unsymmetrical subunits, which were grouped into two different stereochemistries. The colletodiol-type compound possessed a 6*R*,14*R* configuration, whereas the clonostachydiol group adopted the opposite orientation (Appendix A). Additionally, these compounds had different skeletal number notations in the 14-membered ring.

Halosmysin A (**1**), which had a unique skeleton and possessed a thiosilvatin analogue conjugated to the 14-membered macrodiolide, might provide further information about the biological activity of this family, as well as its associated biosynthetic pathways. Herein, the structural determination of **1** was identified using ^1^H and ^13^C NMR spectroscopic techniques, while its stereochemistry was defined using NOESY data, information garnered from the *J* values in the ^1^H NMR spectrum, and a plausible biosynthesis from **2**. Since the elucidated stereochemistry of **1** was the same as that of **2**, we adopted the number notations of the 14-membered ring in colletodiol. Additionally, the cytotoxicity of these compounds was tested against murine P388 leukemia, human HL-60 leukemia, and murine L1210 leukemia cell lines.

## 2. Results and Discussion

This fungus was cultured in artificial seawater (80 L) for four weeks. After incubation, the EtOAc extract of the culture filtrate was purified via cytotoxic assay-directed fractionation using a silica gel column, followed by reverse-phase HPLC, affording halosmysin A (**1**) (3.2 mg, 0.11%) and colletodiol (**2**) (20.7 mg, 0.69%). Since the spectral data, including the specific rotation of **2**, were consistent with published reports [6,10], **2** was identified as (6*R*,11*R*,12*R*,14*R*)-colletodiol. This section may be divided by subheadings. It should provide a concise and precise description of the experimental results, their interpretation, and the experimental conclusions which can be drawn therefrom.

Halosmysin A (**1**) was assigned the formula C_31_H_38_N_2_O_9_S using HRFABMS *m*/*z* 615.2380 [M + H]^+^ (calcd for C_31_H_39_N_2_O_9_S: 615.2376) (Appendix A). Absorption in the IR spectrum at 3393, 2930, 2862, 1752, 1706, 1657, and 1612 cm^−1^ indicated the presence of hydroxy, amido, and ester moieties, as well as an aromatic ring. Close inspection of the ^1^H and ^13^C NMR spectra (Table 1 and Appendix A) of **1** using distortionless enhancement by polarization transfer (DEPT) and ^1^H–^13^C correlation spectroscopy (HSQC), revealed the presence of: two secondary methyls (C-15 and C-16); two olefinic methyls (C-15’ and C-16’); one thiomethyl group (3’-SCH_3_); four sp^3^-hybridized methylenes (C-5, C-13, C-7’, and C-12’), of which one was an oxygen-bearing sp^3^-methine (C-12’); five sp^3^-methines (C-6, C-9, C-10, C-12, and C-14), of which three were oxygen-bearing sp^3^-methines (C-6, C-12, and C-14); seven sp^2^-methines (C-3, C-4, C-9’, C-10’, and C-13’); two quaternary sp^3^-carbons (C-3’, and C-6’); three quaternary sp^2^-carbons (C-8’, C-11’, and C-14’); and five carbonyl groups (C-2, C-8, C-11, C-2’, and C-5’). In ^1^H–^1^H correlation spectroscopy (COSY), the correlations indicated in bold lines were observed, and five partial structures were elucidated (Figure 2). Analysis of the HMBC spectrum showed the connection of these units with the remaining functional groups. The correlations from H-4 to C-2, from H-9 to C-8 and C-11, from H-10 to C-8, and from H-13 to C-11, confirmed the presence of the 14-membered bislactone moiety together with the ^1^H NMR chemical shifts at H-6 and H-14 (*δ* ppm 5.30 and 5.23, respectively) (Figure 2, Appendix A). The large coupling constant (*J*_3,4_ = 16.2 Hz) indicated that the geometrical configuration between C-3 and C-4 was *E*. This bislactone had the same planar structure as colletoketol [6] except for the saturated olefin between C-9 and C-10 (Appendix A). On the other hand, the HMBC correlations—from H-1’ (NH) to C-3’, from H-4’ (NH) to C-2’ and C-6’, from S-CH_3_ to C-3’, from H-7’ to C-5’, C-6’, C-8’, and C-9’, from H-10’ to C-8’ and C-11’, from H-12’ to C-11’ and C-14’, and from H-15’ and H-16’ to C-13’ and C-14’—confirmed the presence of the thiosilvatin analogue (Figure 2, Appendix A), which had been previously isolated from both *Tolypocladium* sp. [24] and *Fusarium* sp. [25,26] as the 3,6-bis(methylthio)-2,5-piperazinedione derivative [27]. Additionally, the correlations from H-9 to C-6’, from H-10 to C-3’, and from H-7’ to C-9, showed that the 14-membered bislactone moiety and the thiosilvatin analogue were bound between C-9 and C-6’ and between C-10 and C-3’ (Figure 2, Appendix A), thereby establishing the planar structure of **1**, as shown in Figure 1.

For the stereochemistry of **1**, both the relative configuration and the conformation were investigated via nuclear Overhauser effect spectroscopy (NOESY) experiments and the ^1^H NMR coupling constant (Appendix A and Figure 3). All of the attempts to induce the crystallization of **1** were unsuccessful. For the 14-membered ring moiety of **1**, the NOESY correlations between H-3 and H-5 and those between H-4 and H-6, as well as the large coupling constants (*J*_5__α,6_ = 13.2 Hz, *J*_5__β,4_ = 10.2 Hz), confirmed the presence of dihedral angles between C-6 and C-5 and between C-5 and C-4 (Figure 3; Newman projection formula A). On the other hand, the NOESY correlations from H-16 to H-13 and H-13, and the large coupling constants (*J*_13__β,14_ = 12.0 Hz), revealed the dihedral angle between C-14 and C-13 (Figure 3; Newman projection formula B). The NOESY correlations between H-3 and H-9, between H-9 and H-4’ (NH), and between H-10 and H-4’ (NH), showed that both H-9 and H-10 were oriented to the configuration. Additionally, the orientation for H-12 was deduced from the NOESY correlations between H-13 and H-10, and those between H-12 and 3’-SCH_3_, together with the small coupling constants (*J*_13__α,12_ = 7.8 Hz, *J*_13__β,12_ = 1.2 Hz). In the thiosilvatin analogue moiety, the NOESY correlations between H-4’ (NH) and H-9, H-10, and 3’-SCH_3_, between H-12 and 3’-SCH_3_, between H-1’ (NH) and H-7’ and H-9’, and between H-7’ and H-9 and H-15, revealed the stereochemistry of C-3’ and C-6’. Based on the evidence presented, the relative configuration of **1** was established (Figure 1), which was the same as that of the other major co-metabolite from this fungus, colletodiol (**2**). Based on the information obtained from past synthetic studies about the stereochemistry of this class of 14-membered bislactones, we assumed that **1** belonged to the family of colletoketol, colletodiol (**2**), and grahamimycin B in terms of its absolute configuration, i.e., it possessed 6*R*,12*R*,14*R* (Appendix A) [10,12,13,15,19,20,21,22,23]. Confirmation of the absolute configuration of **1** was obtained through alkaline hydrolysis. As expected, the reaction produced (–)-5-hydroxy-(2*E*)-hexenoic acid, which corresponded to the triketide moiety of the same family of macrodiolides; confirmation of this was obtained from the ^1^H NMR spectrum and its specific rotation [6,10].

Previous biosynthetic studies showed that colletodiol (**2**) was produced from the epoxide derivative (I) induced from the cyclic triene (II), which was formed via cyclization between the C_6_ precursor (III) and the C_8_ precursor (IV) [28,29]. Additionally, oxygen-labeling experiments revealed that colletoketol was derived from **2** via the oxidative process [28]. Therefore, we hypothesized the biosynthetic pathway of **1** from colletoketol. Briefly, the 6-SCH_3_ group in the 3,6-bis(methylthio)-2,5-piperazinedione derivative (the thiosilvatin analogue) was eliminated as a result of the abstraction of an amide proton via enzymatic dehydrogenation. Subsequently, the *π*-electron at C-9 in the macrodiolide attacked C-6 in the thiosilvatin analogue as a nucleophile, forming a bond between C-9 and C-6’ in **1**. Next, C-10 in the macrodiolide was attacked by C-3 in the piperazinedione, which eliminated a proton to form the bond between C-10 and C-3’ in **1** (Scheme 1). Since our hypothesis of the biosynthetic pathway suggests that stereoisomers of **1** at C-9, C-10, C-3’, or C-6’ exist, we will conduct further research to isolate and characterize the related metabolites from this fungus.

The cancer cell growth inhibitory properties of halosmysin A (**1**) and colletodiol (**2**) were examined using murine P388 leukemia, human HL-60 leukemia, and murine L1210 leukemia cell lines as a primary screen for antitumor activity (Table 2). According to the results, **1** exhibited a potency on par with that of 5-fluorouracil against the above-mentioned cell lines, whereas **2** did not inhibit cell growth at all. In a previous paper [25], the cytotoxicity of the thiosilvatin analogue was reported to be weak or inactive. Interestingly, the compound conjugated between the macrodiolide and the piperazinedione derivatives such as **1**, significantly inhibited the growth of cancer cell lines. Since there are no reports on macrodiolide bound to other molecules, the study of the structure–activity relationship is yet to begin. In order to examine the structure–activity relationship and elucidate details about the compound’s mode of action, we will conduct further studies on related fungal metabolites, including the stereoisomers of **1**.

## 3. Materials and Methods

### 3.1. General Experimental Procedures

The following procedures are the same as those in previous reports [5]. The NMR spectra were recorded on an Agilent-NMR-vnmrs (Agilent Technologies, Santa Clara, CA, USA) 600 with tetramethylsilane (TMS) as an internal reference. FABMS was recorded using a JEOL JMS-7000 mass spectrometer (JEOL, Tokyo, Japan). IR spectra were recorded on a JASCO FT/IR-680 Plus (Tokyo, Japan). Optical rotations were measured using a JASCO DIP-1000 digital polarimeter (Tokyo, Japan). Silica gel 60 (230–400 mesh, Nacalai Tesque, Inc., Kyoto, Japan) was used for column chromatography with medium pressure. ODS HPLC was run on a JASCO PU-1586 (Tokyo, Japan), equipped with a differential refractometer RI-1531 (Tokyo, Japan) and Cosmosil Packed Column 5C18-MSII (25 cm × 20 mm i.d., Nacalai Tesque, Inc., Kyoto, Japan). Analytical TLC was performed on precoated Merck aluminum sheets (DC-Alufolien Kieselgel 60 F254, 0.2 mm, Merck, Darmstadt, Germany) with the solvent system CH_2_Cl_2_–MeOH (19:1) (Nacalai Tesque, Inc., Kyoto, Japan), and compounds were viewed under a UV lamp (AS ONE Co., Ltd., Osaka, Japan) and sprayed with 10% H_2_SO_4_ (Nacalai Tesque, Inc., Kyoto, Japan), followed by heating.

### 3.2. Fungal Material

The fungus *Halosphaeriaceae* sp. was isolated from the surface of the marine alga *Sargassum thunbergii. Halosphaeriaceae* sp. was collected at Osaka Bay, Japan in July 2017. The fungal strain was identified based on the result of its internal transcribed spacer (ITS) rDNA nucleotide sequence analysis by Techno Suruga Laboratory Co., Ltd. (Shizuoka, Japan). The alga *Sargassum thunbergii* was wiped with EtOH and a cutting applied to the surface of the nutrient agar layered in a Petri dish. Serial transfers of one of the resulting colonies provided a pure strain of *Halosphaeriaceae* sp.

### 3.3. Culturing and Isolation of Metabolites

The fungus was cultured at 27 °C for four weeks in a medium (80 L) containing 1% glucose, 1% malt extract, and 0.05% peptone in artificial seawater adjusted to pH 7.5. Then, the culture filtrate was extracted thrice with AcOEt. The combined extracts were evaporated in vacuo to afford a mixture of crude metabolites (3.0 g). The EtOAc extract was chromatographed on a silica gel column, with a CH_2_Cl_2_/MeOH gradient as the eluent to afford Fr. 1 (2% MeOH in CH_2_Cl_2_ eluate, 121.2 mg). Fr. 1 was purified by ODS HPLC, using MeOH/H_2_O (70:30) as the eluent to afford **1** (3.2 mg) and **2** (20.7 mg). 

Halosmysin A (**1**): pale yellow oil; ${\left[\alpha \right]}_{22}^{D}$ +105.6 (*c* 0.048, CHCl_3_); IR (neat) *ν*_max_ / cm^−1^: 3193, 2930, 2862, 1722, 1706, 1657, 1612, 1510. HRFABMS *m*/*z* 615.2380 [M + H]^+^ (calcd for C_31_H_39_N_2_O_9_S: 615.2376); NMR data, see Table 1 and Appendix A.

### 3.4. Alkaline Hydrolysis of ***1***

First, **1** (2.8 mg) and 0.1 N NaOH aq. (1 mL) were stirred at room temperature for 15 h, then acidified with 1 N HCl and extracted with AcOEt. The organic layer was evaporated in vacuo, and the residue was purified by ODS HPLC, using MeOH/H_2_O (0.1% AcOH) (30:70) as the eluent to (–)-5-hydroxy-(2E)-hexenoic acid (0.9 mg) (r.t. 28.5 min). 

(–)-5-hydroxy-(2E)-hexenoic acid: clear oil; ${\left[\alpha \right]}_{22}^{D}$ -14.3 (c 0.09, EtOH); ^1^H NMR (600 MHz, MeOH-d_4_) δ ppm: 1.16 (3H, d, J = 6.0 Hz), 2.31 (2H, dd, J = 6.0, 6.0 Hz), 3.85 (1H, sext, J = 6.0 Hz), 5.92 (1H, br s), 6.88 (1H, m).

### 3.5. Assay for Cytotoxicity

The cytotoxic activities of **1** and **2** were examined by the same procedure to date [5]—the 3-(4,5-dimethyl-2-thiazolyl)-2,5-diphenyl- 2H-tetrazolium bromide (MTT) method. The P388, HL-60, and L1210 cells were cultured in an RPMI 1640 Medium (10% fetal calf serum) at 37 °C in 5% CO_2_. The test materials were dissolved in dimethyl sulfoxide (DMSO) to give a concentration of 10 mM, and the solution was diluted with the Essential Medium to yield concentrations of 200, 20, and 2 M, respectively. Each solution was combined with each cell suspension (1 × 10^−5^ cells/mL) in the medium, respectively. After incubating at 37 °C for 72 h in 5% CO_2_, the grown cells were labeled with 5 mg/mL MTT in phosphate-buffered saline (PBS), and the absorbance of formazan dissolved in 20% sodium dodecyl sulfate (SDS) in 0.1 N HCl, was measured at 540 nm with a microplate reader (MTP-310, CORONA electric). Each absorbance value was expressed as a percentage relative to that of the control cell suspension, which was prepared without the test substance, using the same procedure described above. All of the assays were performed three times, semilogarithmic plots were constructed from the averaged data, and the effective dose of the substance required to inhibit cell growth by 50% (IC_50_) was determined.

### 3.6. The Origin of the Cell Lines

The P388 cell line was obtained from Dr. Numata, the HL-60 cell line from Dr. Kawai (death), and the L1210 cell line from Dr. Endo.

## 4. Conclusions

In conclusion, we have isolated a new macrodiolide, halosmysin A (**1**), as a structurally unprecedented secondary metabolite from the fungus *Halosphaeriaceae* sp., separated from the marine alga *Sargassum thunbergii.* In addition, we have proposed a plausible biosynthetic pathway toward **1** from the known macrodiolide, colletodiol (**2**), which was also isolated from the same fungal strain. This compound exhibited stronger cytotoxicity than **2** and the piperazinedione derivative, which were constituent molecules of **1**. In the near future, we will publish further research on the new, fused macrodiolides or the stereoisomer of **1**, to elucidate both the structure–activity relationship and the mechanism of activity.

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
