# Peer review of "Halosmysin A, a Novel 14-Membered Macrodiolide Isolated from the Marine-Algae-Derived Fungus Halosphaeriaceae sp."

_marinedrugs, 2020, doi:10.3390/md18060320_

Round 1

Reviewer 1 Report

In  Halosmysin A, a novel 14-membered macrodiolide isolated from the marine-algae-derived fungusHalosphaeriaceae sp. Yamada and co-authors reported of isolation and characterization of a new 14-membered macrodiolide with an unprecedented skeleton from fungus Halosphaeriaceae sp. associate to marine algae Sargassum thunbergii. The antiproliferative activity against different tumoral cell lines was also evaluated.

The novelty of an original chemical structure could be interesting, but unfortunately the manuscript is lacking in some areas as reported below.

  1. The sentence at lines 25-26 “Our group has published numerous papers featuring exploratory research on marine-derived fungal metabolites to date [5–8]” and related references are inappropriate self-citations.

  1. The second unit of halosmysin A is called “diketopiperazine derivate”. In my opinion it should be called thiosilvatin analogue. Please refer to Salvatore, M. M., Nicoletti, R., DellaGreca, M., & Andolfi, A. (2019). Occurrence and Properties of Thiosilvatins. Marine drugs, 17(12), 664.

  1. At line 68, change “C31H38N2O9S” into “C31H39N2O9S”

  1. The absolute configuration (6R, 12R, 14R) of the asymmetric carbons of the macrodiolidic ring was assigned by Noesy spectra and by analogy with what was observed for colletoketol, colletodiol and grahamimycin B. In particular, about the sentence reported on line 117 " Confirmation of the absolute configuration of 1 was obtained through alkaline hydrolysis. " no information is given on the procedure used for the reaction. Quantity of 1 used, times, concentrations, work-up of reaction, purification method of hydrolysis products. How was (-) - 5-hydroxy- (2E) -hexenoic acid separated? Was the second unit of the macrodiolide ring also obtained?

  1. No information is given about the stereochemistry of C-3 'and C-6' of the thiosilvatin moiety

  1. Information on grow condition is reported in section Results and Discussion (Lines 56-58) while in section 3. Materials and Methods (line 173) the authors reported “This fungus was cultured in artificial seawater (80 L) for four weeks ". Please remove the information from paragraph 2 and insert it in paragraph 3. There is no information on extract procedures of culture filtrates. Please added some information about it.

  1. The identification of fungus was performed by an external laboratory (line 169) but the authors don’t give information about the method used. Please enter this information, moreover the identification number of the strain is reported only in the abstract (line 12).

  1. About the sentence at line 143 “the cytotoxicity of the piperazinedione derivatives was reported to be weak or inactive.” Please see comment at point 2).

  1. Information about other secondary metabolites could be of interest to the scientific community. Besides 2, have not been identified other compounds?

Author Response

Reviewer #1

Halosmysin A, a novel 14-membered macrodiolide isolated from the marine-algae-derived fungusHalosphaeriaceae sp. Yamada and co-authors reported of isolation and characterization of a new 14-membered macrodiolide with an unprecedented skeleton from fungus Halosphaeriaceae sp. associate to marine algae Sargassum thunbergii. The antiproliferative activity against different tumoral cell lines was also evaluated.

The novelty of an original chemical structure could be interesting, but unfortunately the manuscript is lacking in some areas as reported below.

1.The sentence at lines 25-26 “Our group has published numerous papers featuring exploratory research on marine-derived fungal metabolites to date [5–8]” and related references are inappropriate self-citations.

Response: As be pointed out, we removed a part of self-citation. But we would like to leave number 5 as an example.

2.The second unit of halosmysin A is called “diketopiperazine derivate”. In my opinion it should be called thiosilvatin analogue. Please refer to Salvatore, M. M., Nicoletti, R., DellaGreca, M., & Andolfi, A. (2019). Occurrence and Properties of Thiosilvatins. Marine drugs, 17(12), 664.

Response: Thank you for your opinion and presentation of reference. As be your pointed out, we replaced “diketopiperazine derivative” to “thiosilvatin analogue” throughout the text, and added the paper you presented us as a new reference number 27.

3.At line 68, change “C31H38N2O9S” into “C31H39N2O9S”

Response: Thank you for your pointed out, we changed “C31H38N2O9S” into “C31H39N2O9S”

4.The absolute configuration (6R, 12R, 14R) of the asymmetric carbons of the macrodiolidic ring was assigned by Noesy spectra and by analogy with what was observed for colletoketol, colletodiol and grahamimycin B. In particular, about the sentence reported on line 117 " Confirmation of the absolute configuration of 1 was obtained through alkaline hydrolysis. " no information is given on the procedure used for the reaction. Quantity of 1 used, times, concentrations, work-up of reaction, purification method of hydrolysis products. How was (-) - 5-hydroxy- (2E) -hexenoic acid separated? Was the second unit of the macrodiolide ring also obtained?

Response: We added the description for the procedure of the alkaline hydrolysis in section Materials and Methods as section number 3.4. The second unit of the macrodiolide ring has not been obtained at this time.

5.No information is given about the stereochemistry of C-3 'and C-6' of the thiosilvatin moiety

Response: To around line 110, we added the description for the stereochemistry of C-3 'and C-6'. It showed NOESY correlation necessary to determine the stereochemistry of C-3 'and C-6'.

6.Information on grow condition is reported in section Results and Discussion (Lines 56-58) while in section 3. Materials and Methods (line 173) the authors reported “This fungus was cultured in artificial seawater (80 L) for four weeks ". Please remove the information from paragraph 2 and insert it in paragraph 3. There is no information on extract procedures of culture filtrates. Please added some information about it.

Response: Thank you for your pointed out, we removed the information on grow condition in section Results and Discussion, and insert it in section Materials and Methods. We added the information on extract procedures of culture filtrates newly in section Materials and Methods.

7.The identification of fungus was performed by an external laboratory (line 169) but the authors don’t give information about the method used. Please enter this information, moreover the identification number of the strain is reported only in the abstract (line 12).

Response: We added the information for the method of the fungal identification to sention Materials and Methods 3.2.

8.About the sentence at line 143 “the cytotoxicity of the piperazinedione derivatives was reported to be weak or inactive.” Please see comment at point 2).

Response: As be pointed out, we corrected it by referring to the comments 2.

9.Information about other secondary metabolites could be of interest to the scientific community. Besides 2, have not been identified other compounds?

Response: Thank you for your comment. Recently, we have detected the existence of other analogs, but the amount is too small to determine its structure. We are currently investigating supplementation of the compound by repeating the culture.

Reviewer 2 Report

This an excellent paper dealing with the isolation and identification of a new macrodiolide. The structure is well assigned and a plausible biosynthetic route is proposed. Biological activity studies on three different cell lines complement this study.

I strongly favor it’s acceptance.

Author Response

Reviewer #2

General remarks

This an excellent paper dealing with the isolation and identification of a new macrodiolide. The structure is well assigned and a plausible biosynthetic route is proposed. Biological activity studies on three different cell lines complement this study.

I strongly favor it’s acceptance.

English suggestions:

Instead of ‘---‘ – use ‘---'

Line 15 – ‘derivatisation and the comparison’ – ‘derivatisation and comparison’

Response: Thank you for your pointed out, we changed ‘derivatisation and the comparison’ to ‘derivatisation and comparison’.

Line 47 – ‘might provid further’ – ‘might provide further’

Response: Thank you for your pointed out, we corrected ‘might provid further’ to ‘might provide further’.

Line 198 – ‘We have isolated new macrodiolide’ – ‘We have isolated a new macrodiolide’

Response: Thank you for your pointed out, we corrected ‘We have isolated new macrodiolide’ to‘We have isolated a new macrodiolide’.

  1. Introduction

No corrections

  1. Results and discussion

Figure 1 – Correct legend to macrodiolides

Response: Thank you for your pointed out, we corrected legend to macrodiolides.

Table 1 – Insert frequency in legend. Complete footnote as in table S1

Response: As be pointed out, we inserted frequency to footnote, and complete footnote for 1H NMR as in table S1.

Figure 2 – correct to ‘.correlations for 1’

Response: Thank you for your pointed out, we corrected legend of Figure 2 to ‘correlations for 1’

Line 84 – correct ‘trans’ – to ‘E’

Response: As be pointed out, we corrected ‘trans’ to ‘E’

Lines 98/109 – there are some missing ‘alphas’ and ‘betas’

Response: Thank you for your pointed out, we added ‘alphas’ and ‘betas’ to methylene proton H-13.

HMBC correlations

Lines 79/82 – I believe you mean correlations from H4 to C2 and H13 to C11 (according to Figure 2).

Response: Thank you for reviewing the details. As be pointed out, we correct NOESY correlations.

  1. Materials and Methods

3.1 – no corrections

3.2 – no corrections

3.3 – no corrections

3.4 - no corrections

3.5 - no corrections

  1. Conclusions:

No corrections

  1. References

The title is missing in Ref. 16

Response: Thank you for your pointed out, we added a title to Ref. 16.

Supplementary material

Please include IR spectra of compound 1 in word file. Correct the index accordingly.

Table S1 – insert frequency in legend.

Response: As be pointed out, we added IR spectra of compound 1 in SM.

         As be pointed out, we inserted frequency to footnote of Table S1.

Reviewer 3 Report

The authors reported isolation and structure establishing of a new natural product, Halosmysin A, a novel 14-membered macrodiolide. The structure of this compound, including configuration, has been determined by spectroscopic methods, especially by NMR 2D techniques (COSY, HMBC, NOESY). Then, the authors presented a plausible biosynthetic pathway toward the title compound. I have no objections to the presentation of the material, although the description of the NMR results analysis procedure is quite difficult to follow. I recommend publishing this work as it is, without amendments.

Author Response

Reviewer #3

The authors reported isolation and structure establishing of a new natural product, Halosmysin A, a novel 14-membered macrodiolide. The structure of this compound, including configuration, has been determined by spectroscopic methods, especially by NMR 2D techniques (COSY, HMBC, NOESY). Then, the authors presented a plausible biosynthetic pathway toward the title compound. I have no objections to the presentation of the material, although the description of the NMR results analysis procedure is quite difficult to follow. I recommend publishing this work as it is, without amendments.

Response: Thank you for your evaluation.

Since the conformational movement of the compound was stable, NMR analyses could prove the stereochemistry of the compound to some extent. We speculate that the large functional group such as diketopiperazine led to this stabilization. We will further promote the isolation of analogs and will report them in the near future.

Round 2

Reviewer 1 Report

The manuscript ‘Halosmysin A, a novel 14-membered macrodiolide isolated from the marine-algae-derived fungus Halosphaeriaceae sp’ is accepted for publication in current form. The authors answered all comments and the manuscript improved after the review process.